# ADVERSARIAL SPHERES

## ABSTRACT

State of the art computer vision models have been shown to be vulnerable to small adversarial perturbations of the input. In other words, most images in the data distribution are both correctly classified by the model and are very close to a visually similar misclassified image. Despite substantial research interest, the cause of the phenomenon is still poorly understood and remains unsolved. We hypothesize that this counter intuitive behavior is a naturally occurring result of the high dimensional geometry of the data manifold. As a first step towards exploring this hypothesis, we study a simple synthetic dataset of classifying between two concentric high dimensional spheres. For this dataset we show a fundamental tradeoff between the amount of test error and the average distance to nearest error. In particular, we prove that *any* model which misclassifies a small constant fraction of a sphere will be vulnerable to adversarial perturbations of size $O(1/\sqrt{d})$. Surprisingly, when we train several different architectures on this dataset, all of their error sets naturally approach this theoretical bound. As a result of the theory, the vulnerability of neural networks to small adversarial perturbations is a logical consequence of the amount of test error observed. We hope that our theoretical analysis of this very simple case will point the way forward to explore how the geometry of complex real-world data sets leads to adversarial examples.

## 1 INTRODUCTION

There has been substantial work demonstrating that standard image models exhibit the following phenomenon: most randomly chosen images from the data distribution are correctly classified and yet are close to a visually similar nearby image which is incorrectly classified (Goodfellow et al., 2014; Szegedy et al., 2014). This is often referred to as the phenomenon of *adversarial examples*. These adversarially found errors can be constructed to be surprisingly robust, invariant to viewpoint, orientation and scale (Athalye & Sutskever, 2017). Despite some theoretical work and many proposed defense strategies (Cisse et al., 2017; Madry et al., 2017; Papernot et al., 2016) the cause of this phenomenon is still poorly understood.

There have been several hypotheses proposed regarding the cause of adversarial examples. We briefly survey some of them here. One common hypothesis is that neural network classifiers are too linear in various regions of the input space, (Goodfellow et al., 2014; Luo et al., 2015). Another hypothesis is that adversarial examples are off the data manifold (Goodfellow et al., 2016; Anonymous, 2018b;a; Lee et al., 2017). Cisse et al. (2017) argue that large singular values of internal weight matrices may cause the classifier to be vulnerable to small perturbations of the input.

Alongside works endeavoring to explain adversarial examples, others have proposed defenses in order to increase robustness. Some works increase robustness to small perturbations by changing the non-linearities used (Krotov & Hopfield, 2017), distilling a large network into a small network (Papernot et al., 2016), or using regularization (Cisse et al., 2017). Other works explore detecting adversarial examples using a second statistical model (Feinman et al., 2017; Abbasi & Gagné, 2017; Grosse et al., 2017; Metzen et al., 2017). However, many of these methods have been shown to fail (Carlini & Wagner, 2017a;b). Finally, adversarial training has been shown in many instances to increase robustness (Madry et al., 2017; Kurakin et al., 2016; Szegedy et al., 2014; Goodfellow et al., 2014). Despite some progress on increasing robustness to adversarial perturbations, local errors have still been shown to appear for distances just beyond what is adversarially trained for Sharma & Chen (2017).

This phenomenon is quite intriguing given that these models are highly accurate on the test set. We hypothesize that this behavior is a naturally occurring result of the high dimensional nature of the data manifold. In order to begin to investigate this hypothesis, we define a simple synthetic task of classifying between two concentric high dimensional spheres. This allows us to study adversarial examples in a setting where the data manifold is well defined mathematically and where we have an analytic characterization of the decision boundary learned by the model. Even more importantly, we can naturally vary the dimension of the data manifold and study the effect of the input dimension on the geometry of the generalization error of neural networks. Our experiments and theoretical analysis on this dataset demonstrate the following:

- A similar behavior to that of image models occurs: most randomly chosen points from the data distribution are correctly classified and yet are "close" to an incorrectly classified input. This behavior occurs even when the test error rate is less than 1 in 10 million.
- For this dataset, there is a fundamental tradeoff between the amount of generalization error and the average distance to the nearest error. In particular, we show that *any* model which misclassifies a small constant fraction of the sphere will be vulnerable to adversarial perturbations of size $O(1\sqrt{d})$.
- Neural networks trained on this dataset naturally approach this theoretical optimal tradeoff between the measure of the error set and the average distance to nearest error. This implies that in order to linearly increase the average distance to nearest error, the error rate of the model must decrease exponentially.
- We also show that models trained on this dataset may become extremely accurate even when ignoring a large fraction of the input.

We conclude with a detailed discussion about the connection between adversarial examples for the sphere and those for image models.

## 2 THE CONCENTRIC SPHERES DATASET

The data distribution is mathematically described as two concentric spheres in $d$ dimensions: we generate a random $x \in \mathbb{R}^d$ where $||x||_2$ is either $1.0$ or $R$, with equal probability assigned to each norm (for this work we choose $R = 1.3$). We associate with each $x$ a target $y$ such that $y = 0$ if $||x||_2 = 1$ and $y = 1$ if $||x||_2 = R$.

Studying a synthetic high dimensional dataset has many advantages:

- The probability density of the data $p(x)$ is well defined and is uniform over all $x$ in the support. We can also sample uniformly from $p(x)$ by sampling $z \sim N(\vec{0}, I)$ and then setting $x = z/||z||_2$ or $x = Rz/||z||_2$.
- There is a theoretical max margin boundary which perfectly separates the two classes (the sphere with radius $(R + 1)/2$).
- We can design machine learning models which provably can learn a decision boundary which perfectly separate the two spheres.
- We can control the difficulty of the problem by varying $d$, and $R$.

Our choice of $R = 1.3$ was a bit arbitrary and we did not explore in detail the relationship between adversarial examples and the distance between to two spheres. Additionally, our choice to restrict the data distribution to be the shells of the two spheres was made to simplify the problem further.

In our experiments we investigate training on this dataset in two regimes. First, the online setting where each minibatch is a uniform sample from $p(x)$ ($N = \infty$). Second, where there is a fixed training set of size $N$ and the network is trained for many epochs on this finite sample.

## 3 ADVERSARIAL EXAMPLES FOR A DEEP ReLU NETWORK

Our first experiment used an input dimensionality of $d = 500$. We then train a 2 hidden layer ReLU network with 1000 hidden units on this dataset. We applied batch normalization (Ioffe & Szegedy,

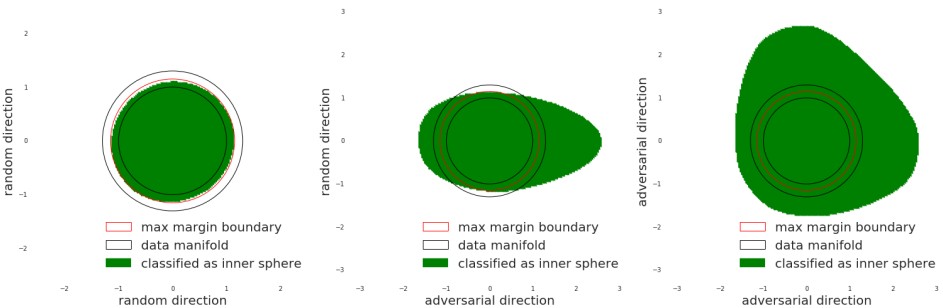

Figure 1: Visualizing a 2d slice of the input space where the subspace is spanned by: 2 randomly chosen directions (**left**), 1 random and 1 "adversarial direction" (**center**), and 2 orthogonal "adversarial directions" (**right**). The data manifold is indicated in black and the max margin boundary in red. The green area indicates points which are classified by the ReLU network as being on the inner sphere. In the last plot, the projection of the entire outer sphere is misclassified despite the fact that the error rate of the model is less than 1 in 10 million.

2015) to the two hidden layers, but not to the readout layer. We train with minibatch SGD, minimizing the sigmoid cross entropy loss. We use Adam optimizer (Kingma & Ba, 2014) for 1 million training steps with mini batch size 50 and learning rate 0.0001. Because this is training in the online setting with batch size 50 and 1 million training points, 50 million data points were used during training.

We evaluated the final model on 10 million uniform samples from each sphere - 20 million points in total and observed no errors on these finite samples. Thus the error rate of this model is unknown, we only have a statistical upper bound on the error rate. Despite this, we are able to adversarially find errors *on the data manifold* by performing gradient descent on the spheres (see Section 3.1). There are two types of adversarial examples we generate using this method, the first are *worst-case examples*, where we iterate the attack until the attack objective converges and do not restrict to a local region around the starting point. The second type are *nearest neighbor examples*, where we terminate the attack on the first misclassification found.

In Figure 1 we visualize the decision boundary by taking different 2d projections of the 500 dimensional space. When we take a random projection, the model has closely approximated the max margin boundary on this projection. Note the model naturally interpolates between the two spheres despite only being trained on samples from the surfaces of the spheres. By contrast, when we take a 2d projection where one basis vector is a worst-case adversarial example, the model's decision boundary is highly warped along this "adversarial direction". There are points of norm > 2 for which the model is confident is on the inner sphere. We can also take a slice where the x and y axis are an orthogonal basis for the subspace spanned to two different worst-case examples. Although the last plot shows that the entire projection of the outer sphere is misclassified, the volume of this error region is exceedingly small due to the high dimensional space.

Despite being extremely rare, these misclassifications appear close to randomly sampled points on the sphere. The mean L2 distance to the nearest error on the data manifold is 0.18, by comparison two randomly sampled points on the inner sphere are typically around $\sqrt{2} \approx 1.41$ distance from each other. If we look for the nearest point in between the two spheres which is classified as being on the outer sphere, then we get an average L2 distance of 0.0796, and an average norm of 1.07. Thus the nearest example of the other class is typically about half the distance to the theoretical margin.

This phenomenon of very unlikely but local errors appears only when the spheres are high dimensional. In Figure 2 (right), we visualize the same model trained on 100 samples in the case where $d = 2$. The model makes no errors on the data manifold. In our experiments the highest dimension we were able to train the ReLU net where no errors can be found adversarially (local or not) seems to be around $d = 60$.

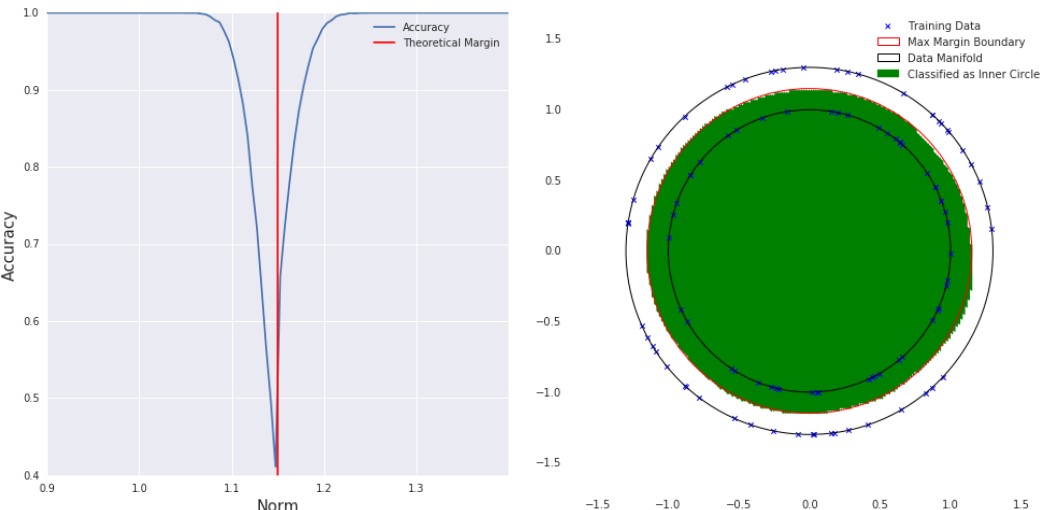

Figure 2: **Left:** We consider the ReLU net trained on 50 million samples from two 500 dimensional spheres of radius 1.0 and 1.3. We evaluate the accuracy of this network on the entire space using a theoretical decision boundary of 1.15. For each norm considered we plot the accuracy among 10000 random samples. We see the accuracy rapidly increases as we move away from the margin. As we move far enough away we no longer observe errors on the random samples. However, we are able to adversarially find errors as far as norms .6 and 2.4. **Right:** We trained the same ReLU net on 100 samples from the data distribution when $d = 2$. By visualizing predictions on a dense subset of the entire space it appears that the model makes no errors on either circle.

### 3.1 FINDING ADVERSARIAL EXAMPLES WITH A MANIFOLD ATTACK

Several recent works have hypothesised that adversarial examples are off the data manifold (Goodfellow et al., 2016; Anonymous, 2018b;a; Lee et al., 2017). We wanted to test if adversarial examples were off the data manifold. To that end we designed an attack which specifically produces adversarial examples on the data manifold which we call a *manifold attack*. Traditional attack methods for image models start with an input $x$ and target class $\hat{y}$ and finds an input $\hat{x}$ that maximizes $P(\hat{y} \mid \hat{x})$ subject to the constraint $||x - \hat{x}|| < \epsilon$, where $|| \cdot ||$ is often chosen to be the $L_\infty$ norm.

The manifold attack maximizes $P(\hat{y} \mid \hat{x})$ subject to the constraint $||\hat{x}||_2 = ||x||_2$. This ensures that the produced adversarial example is of the same class as the starting point and lies in the support of the data distribution. We solve this constraint problem using projected gradient descent (PGD), only for the projection step we project back on the sphere by normalizing $||\hat{x}||_2$. Because this attack only produces adversarial examples on the data manifold, their probability under the data distribution is identical to that of correctly classified points in that $p(x) = p(x_{adv})$.

## 4 ANALYTIC FORMS FOR A SIMPLER NETWORK

It is difficult to reason about the learned decision boundary of the ReLU network. To obtain a more complete understanding of the decision boundary, we next study a simpler model. The network, dubbed "the quadratic network", is a single hidden layer network where the pointwise non-linearity is a quadratic function, $\sigma(x) = x^2$. There is no bias in the hidden layer, and the output simply sums the hidden activations, multiplies by a scalar and adds a bias. With hidden dimension $h$ the network has $d \times h + 2$ learn-able parameters. The logit is written as

$$\hat{y}(x) = w\vec{1}^T (W_1 x)^2 + b \tag{1}$$

where $W_1 \in \mathbb{R}^{h \times d}$, $\vec{1}$ is a column vector of $h$ 1's. Finally, $w$ and $b$ are learned scalars. In the Appendix, we show that the output of this network can be rewritten in the form

$$\hat{y}(x) = \sum_{i=1}^{d} \alpha_i z_i^2 - 1 \tag{2}$$

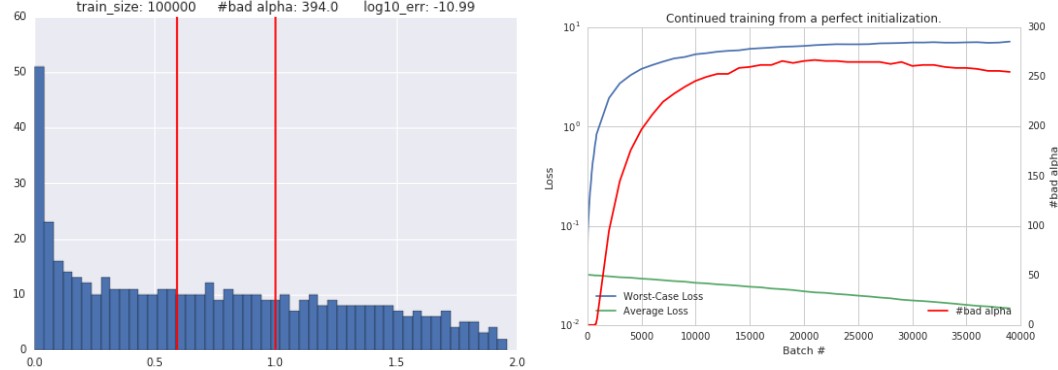

Figure 3: **Left:** The final distribution of $\alpha_i$ when the quadratic network is trained on 100k examples. Red lines indicate the range of needed for perfect classification. Despite a theoretical error rate of 1e-11 most of the $\alpha$ are incorrect. **Right:** Training curves of the quadratic network initialized at a perfect initialization with no classification errors. As training progresses average case loss gets minimized at the cost of a dramatically worse worst case loss. The number of incorrect $\alpha_i$ increases at a similar rate.

where $\alpha_i$ are scalars which depend on the model's parameters and the vector $\vec{z}$ is a rotation of the input vector $\vec{x}$. The decision boundary of the quadratic network is all inputs where $\sum_{i=1}^{d} \alpha_i z_i^2 = 1$. It is an ellipsoid in $d$ dimensions centered at the origin. This allows us to analytically determine when the model has adversarial examples. In particular, if there is any $\alpha_i > 1$, then there are errors on the inner sphere. If there are any $\alpha_i < 1/R^2$ then there are errors on the outer sphere. Therefore, the model has perfect accuracy if and only if all $\alpha_i \in [1/R^2, 1]$.

When we train the quadratic network with $h = 1000$ using the same setup as in Section 3 we arrive at the perfect solution: all of the $\alpha_i \in [1/R^2, 1]$ and there are no adversarial examples. This again was in the online learning setup where each minibatch was an iid sample from the data distribution. The story is different, however, if we train on a finite sample from the data distribution. In particular if we sample $N = 10^6$ data points from $p(x)$ as a fixed finite training set and train using the same setup we arrive at a model which empirically has a very low error rate - randomly sampling 10 million datapoints from each sphere results in no errors, but for which there are adversarial examples. In fact, 394 out of 500 of the learned $\alpha_i$ are incorrect in that $\alpha_i \notin [1/R^2, 1]$ (for a complete histogram see Figure 3).

We can use the Central Limit Theorem (CLT) to estimate the error rate of the quadratic network from the $\alpha_i$ (Section 4.1). The estimated error rate of this particular model to be $\approx 10^{-11}$. Note, we are applying the CLT at the tails of the distribution, so it is unclear how accurate this estimate is. However, we found the CLT closely approximates the error rate in the regime where it is large enough to estimate numerically.

Next we augmented the above setup with a "perfect" initialization; we initialize the quadratic network at a point for which all of the $\alpha_i$ are "correct" but there are non-zero gradients due to the sigmoid cross-entropy loss. The network is initialized at a point where the sigmoid probability of $y = 1$ for the inner sphere and outer spheres is .0016 and 0.9994 respectively. As shown in Figure 3 continued training from this initialization results in a rapid divergence of the worst and average case loss. Although the average loss on the test set decreases with further training, the worst case rapidly increases and adversarial examples can once again be found after 1000 training steps. This behavior results from the fact that the training objective (average sigmoid cross entropy loss) does not directly track the accuracy of the models. It also demonstrates how the worst and average case losses may diverge when the input is high dimensional.

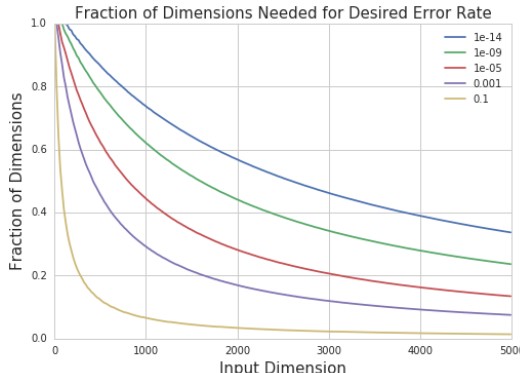

Figure 4: We consider a classification model which only sees a projection of the input, size $d$, onto a $k$ dimensional subspace. We then plot what $k/d$ needs to be in order for the model to obtain a certain error rate. We find that as the input dimension grows, the ratio $k/d$ needed quickly decreases.

### 4.1 ANALYTIC ESTIMATES OF THE ERROR RATE

We can use the CLT[1] to analytically estimate the accuracy for the quadratic network in terms of the $\alpha_i$. The following proposition estimates the error rate on the inner sphere:

**Proposition 4.1** *Consider the decision boundary of the quadratic network of the form*

$$\sum_{i=1}^{d} \alpha_i z_i^2 = 1$$

.

*Let $Z = N(0,1)$. We let $\vec{z} \sim S_0$ to denote that the vector $\vec{z}$ is uniformly distributed on the inner sphere. Finally let $\mu = \sum_{i=1}^{d}(\alpha_i - 1)$ and $\sigma = 2\sum_{i=1}^{d}(\alpha_i - 1)^2$. Then the error rate on the inner sphere can be estimated as*

$$\mathbb{P}_{\vec{z} \sim S_0}\left[\sum_{i=1}^{d} \alpha_i z_i^2 > 1\right] \approx \mathbb{P}_{Z \sim \mathcal{N}(0,1)}\left[Z > -\frac{\mu}{\sigma}\right] = 1 - \Phi\left(\frac{\mu}{\sigma}\right)$$

Proposition 4.1 implies that there are many settings of $\alpha_i$ which obtain very low error rates. As long as $\mathbb{E}[\alpha_i] \approx (1 + R^{-2})/2)$ and their variance is not too high, the model will be extremely accurate. The histogram in Figure 3 illustrates this; i.e. the learned model has an error rate of $10^{-11}$ but 80% of the $\alpha_i$ are incorrect. For a typical sample, the model sums incorrect numbers together and obtains the correct answer. Flexibility in choosing $\alpha_i$ while maintaining good accuracy increases dramatically with the input dimension.

To illustrate this further consider a special case of the quadratic network where the decision boundary is of the form

$$\sum_{i=1}^{k} x_i^2 = b.$$

This simplified model has two parameters, $k$ the number of dimensions the model looks at and $b$ a threshold separating the two classes. How large does $k$ need to be in order for the model to obtain a desired error rate? (Assuming $b$ is chosen optimally based on $k$). We answer this question using the CLT approximation in Proposition 4.1. In Figure 4 we plot the fraction of input dimensions needed to obtain a desired accuracy using this simplified model. For example, if $d = 3000$ then the model can obtain an estimated accuracy of $10^{-14}$ while only looking at 50% of the input.

---

[1]We explored using concentration inequalities to get a true upper bound on the error rate, but found the resulting bounds to be too weak to be of interest. We found the CLT estimate to accurately track the empirical accuracy in the settings where the error rate was large enough to be measured 10 million random samples.

## 5 A SMALL AMOUNT OF CLASSIFICATION ERROR IMPLIES LOCAL ADVERSARIAL EXAMPLES

As demonstrated in section 3, neural networks trained on the sphere dataset exhibit a similar phenomenon to that of image datasets: most random samples from the data distribution are both correctly classified and close to a nearby misclassified point. In this work, we do not attempt to compare the geometry of the natural image manifold to that of the sphere, but we can explain why this property occurs on the sphere dataset.

Let $S_0$ be in the sphere of radius 1 in $d$ dimensions and fix $E \subseteq S_0$ (we interpret $E$ to be the set of points on the inner sphere which are misclassified by some model). For $x \in S_0$ let $d(x, E)$ denote the $L_2$ distance between $x$ and the nearest point in the set $E$. Let $d(E) = \mathbb{E}_{x \sim S_0} d(x, E)$ denote the average distance from a uniformly sampled point on the sphere to the set $E$. Finally, let $\mu(E)$ denote the measure of $E$ as a fraction of the sphere (so $\mu(S_0) = 1$). We prove the following theorem in the Appendix:

**Theorem 5.1** *Consider any model trained on the sphere dataset. Let $p \in [0, 1.0]$ denote the accuracy of the model on the inner sphere, and let $E$ denote the points on the inner sphere the model misclassifies (so in measure $\mu(E) = 1 - p$). Then $d(E) = O(\Phi^{-1}(p)/\sqrt{d})$ where $\Phi^{-1}(x)$ is the inverse normal cdf function.*

This theorem directly links the probability of an error on the test set to the average distance to the nearest error *independently of the model*. Any model which misclassifies a small constant fraction of the sphere must have errors close to most randomly sampled data points, no matter how the model errors are distributed on the sphere. At a high level it follows as direct corollary of an isoperimetric inequality of Figiel et al. (1977). The error set $E$ of fixed measure $\mu(E)$ which maximizes the average distance to the nearest error $d(E)$ is a "cap", which is a set of the form $E = \{x \in S_0 : x_i > \alpha\}$ (or more generally the sphere intersected with a half space). When $d$ is large we can estimate $d(E)$ by using the fact that for $x$ chosen randomly on $S_0$ the distribution of a single coordinate $x_i \approx N(0, 1/d)$. This illustrates the counter-intuitive property of high dimensional spheres, for large $d$ a set of measure say .01% concentrated near a pole will extend all the way to within $O(1/\sqrt{d})$ of the equator, and a randomly chosen $x \sim p(x)$ will with high probability lie close to the equator.

Theorem 5.1 gives an optimal trade off between the amount of generalization error and the average distance to nearest error. We can compare how the error sets of actual trained neural networks compare with this optimal bound. We do this comparison in Figure 5. We train three different architectures on the sphere dataset when $d = 500$, the first is a "small" ReLU network with 1000 hidden units and 2 hidden layers (ReLU-h1000-d2), the second is the quadratic network with 1000 hidden units (Quad-h1000), and the third is a "large" ReLU network with 2000 hidden units and depth 8 (Relu-h2000-d8). We train these networks with varying number of samples from the data distribution, in particular $N \in \{1000, 5000, 10000, 100000, \infty\}$. We then sample the performance of the networks several times during training, computing both the error rate of the model and the average distance to nearest error. The error rate is estimated from 100000 random samples from the data distribution and the average distance is estimated by 100 random points and running PGD for 1000 steps with step size of .001 (searching only on the data manifold for errors). Each point on the plot is a network at a certain snapshot during training (when $N >= 10000$ the networks later in training become so accurate that the error rate cannot be estimated statistically from 100000 samples, these points do not appear in the graph). Surprisingly we see that the trade off between the average distance to nearest error and the amount of error is close to what is optimal if all the errors were concentrated near a "cap" (as represented by a black line). Note that there is some noise in estimating the error rates and average distances (for example PGD is not guaranteed to find the closest error) as a result some networks when sampled appear slightly better than optimal.

This plot suggests that the decision boundaries of these networks are all well behaved given the amount of test error observed. For the quadratic network, the error set on the inner sphere is of the form $E = \{x \in \mathbb{R}^n : ||x||_2 = 1, \sum_{i=1}^{d} \alpha_i x_i^2 > 1\}$. Geometrically this is the area of the sphere which is outside the ellipsoid decision boundary. If $\alpha_i > 1$ for 2 or more $i$, then $E$ is a connected region. This gives some intuition as to why the quadratic network might approach the optimal tradeoff between $d(E)$ and $\mu(E)$. Perhaps more interesting is that both the small and large ReLU networks have similar

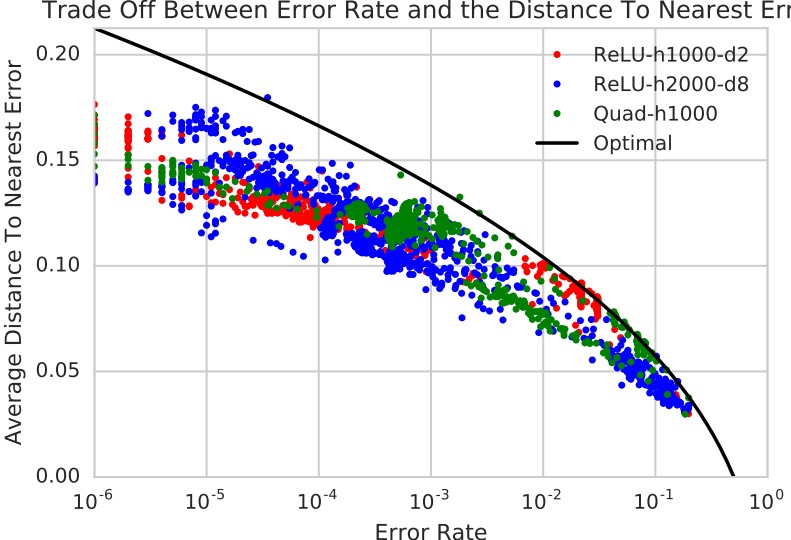

Figure 5: We compare the average distance to nearest error with error rate for 3 networks trained on the sphere dataset. All errors are reported for the inner sphere. The 3 networks are trained with 5 different training set sizes, and their performance are measured at various points during training (the networks eventually become too accurate to appear on this graph, as the error rate will be too small to estimate statistically). Amazingly, we observe that the trade off between the amount of error and the average distance to nearest error closely tracks what is optimal, as would be observed if all the errors were concentrated near a pole of the sphere.

tradeoffs between $d(E)$ and $\mu(E)$, even though they have much more complicated architectures. The large ReLU network, for example, has over 29 million parameters. Despite the complexity, the error region of this large network demonstrates a similar tradeoff as the quadratic network.

# 6 CONCLUSION AND TAKEAWAYS

In this work we attempted to gain insight into the existence of adversarial examples for image models by studying a simpler synthetic dataset. After training different neural network architectures on this dataset we observe a similar phenomenon to that of image models - most random points in the data distribution are both correctly classified and are close to a misclassified point. We then explained this phenomenon for this particular dataset by proving a theoretical tradeoff between the error rate of a model and the average distance to nearest error *independently of the model*. We also observed that several different neural network architectures closely match this theoretical bound.

Theorem 5.1 is significant because it reduces the question of why models are vulnerable to adversarial examples to the question of why is there a small amount of classification error. It is unclear if anything like theorem 5.1 would hold for an image manifold, and future work should investigate if a similar principal applies. Our work suggests that even a small amount of classification error may sometimes logically force the existence of many adversarial examples. This could explain why fixing the adversarial example problem has been so difficult despite substantial research interest. For example, one recent work uses adversarial training to increase robustness in the $L_\infty$ metric (Madry et al., 2017). Although this did increase the size, $\epsilon$, of the perturbation needed to reliably produce an error, local errors still remain for larger $\epsilon$ than those adversarially trained for (Sharma & Chen, 2017).

Several defenses against adversarial examples have been proposed recently which are motivated by the assumption that adversarial examples are off the data manifold (Anonymous, 2018b;a; Lee et al., 2017). Our results challenge whether or not this assumption holds in general. As shown in section 3 there are local errors both on and off the data manifold. Our results raise many questions as to whether or not it is possible to completely solve the adversarial example problem without reducing test error to 0. The test error rate of state of the art image models is non-zero, this implies that a

constant fraction of the data manifold is misclassified and is the unbiased estimate of $\mu(E)$. Perhaps this alone is an indication that local adversarial errors exist.

The concentric spheres dataset is an extremely simple problem which is unlikely to capture all of the complexities of the geometry of a natural image manifold. Thus we cannot reach the same conclusions about the nature of adversarial examples for real-world datasets. However, we hope that the insights gained from this very simple case will point the way forward to explore how complex real-world data sets leads to adversarial examples.

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

# Appendix

## A   PROOF OF THEOREM 5.1

In this section we sketch a proof of Theorem 5.1. Let $E$ be the points on the inner sphere $S_0$ which are misclassified by some model. Let $\mu(E)$ denote the measure of $E$ and $d(E) = \mathbb{E}_{x \sim p} d(x, E)$ denote the average distance from a random point on $S_0$ to the nearest point in $E$.

A "cap" of $S_0$ is the intersection of the sphere with a half space of $\mathbb{R}^d$. Intuitively a cap is all points which are located within some fixed distance of a pole. Figiel et al. (1977) prove that the set $E$ of given measure $\mu(E)$ which maximizes $d(E)$ is a "cap". Thus, without loss of generality we may assume that $E = \{x \in S_0 : x_1 > \alpha/\sqrt{d}\}$ for some $\alpha > 0$. Note that, as $d$ becomes large, the distribution of a single coordinate $x_i$ on the sphere approaches $N(0, \frac{1}{d})$. Thus we have

$$\mathbf{P}[x \in E] \approx \mathbf{P}[N(0, \frac{1}{d}) > \alpha/\sqrt{d}] = \mathbf{P}[N(0, 1) > \alpha].$$

Thus we have $\alpha = \Phi^{-1}(\mu(E))$, because this implies $\mathbf{P}[N(0, \frac{1}{d}) > \alpha/\sqrt{d}] = \mu(E)$.

For $x \in S_0$ let $d(x, E)$ denote the distance from $x$ to the set $E$. This distance is equal to $max(\sqrt{2}(\alpha/\sqrt{d} - x_1), 0)$. Thus we have

$$d(E) = \mathbf{E}_{x \sim S_0} d(x, E) \approx \mathbf{E}\left[max\left(\sqrt{2}(\alpha/\sqrt{d} - N(0, \frac{1}{d})), 0\right)\right] = O(\Phi^{-1}(\mu(E))/\sqrt{d}).$$

## B   THE DECISION BOUNDARY OF THE QUADRATIC NETWORK

Here we show the decision boundary of the quadratic network is a $d$-dimensional ellipsoid. We'll use $\boldsymbol{x}$ to denote the column vector representation of the input. The logit of the quadratic network is of the form

$$l(x) = (W_1\boldsymbol{x})^T(W_1\boldsymbol{x})w + b \tag{3}$$

$W_1$ is the input to hidden matrix, and $w$ and $b$ are scalars. We can greatly simplify this by taking SVD of $W_1 = U\Sigma V^T$. So we have the following:

$$\begin{aligned} l(x) &= (W_1\boldsymbol{x})^T(W_1\boldsymbol{x})w + b \\ &= (U\Sigma V^T\boldsymbol{x})^T(U\Sigma V^T\boldsymbol{x})w + b \\ &= (\boldsymbol{x}^T V\Sigma U^T U\Sigma V^T\boldsymbol{x})w + b \end{aligned}$$

Let $\boldsymbol{z} = V^T\boldsymbol{x}$ which is a rotation of the input. Then the above becomes

$$l(x) = \boldsymbol{z}^T\Sigma^2\boldsymbol{z}w + b \tag{4}$$

Letting the singular values of $W_1$ be the sequence $(s_i)$ we have

$$l(x) = w\sum_{i=1}^{d} s_i^2 z_i^2 + b \tag{5}$$

The decision boundary is of the form $l(x) = 0$, thus we have

$$w \sum_{i=1}^{d} s_i^2 z_i^2 + b = 0$$

$$\sum_{i=1}^{d} \alpha_i z_i^2 - 1 = 0$$

where $\alpha_i = w * s_i^2 / (-b)$.

Note the distribution of $\boldsymbol{z}$ is the same as $\boldsymbol{x}$ (they are rotations of each other) so replacing $z_i$ with $x_i$ above yields a rotation of the decision boundary.

## C  ESTIMATING THE ACCURACY WITH THE CENTRAL LIMIT THEOREM

Here we prove thm. 4.1. As in the statement let $\{\alpha_i\}_{i=1}^{d}$ be nonnegative real numbers and $b > 0$. Let $S_0$ be the unit sphere in $d$-dimensions. Suppose $z$ is chosen uniformly on $S_0$ then we wish to compute the probability that

$$\sum_{i=1}^{d} \alpha_i z_i^2 > 1. \tag{6}$$

One way to generate $z$ uniformly on $S_0$ is to pick $u_i \sim N(0,1)$ for $1 \leq i \leq d$ and let $z_i = u_i / ||u||$. It follows that we may rewrite eq. (6) as,

$$\frac{1}{||u||^2} \sum_{i=1}^{d} \alpha_i u_i^2 > 1$$

$$\sum_{i=1}^{d} \alpha_i u_i^2 > \sum_{i=1}^{d} u_i^2$$

$$\sum_{i=1}^{d} (\alpha_i - 1) u_i^2 > 0. \tag{7}$$

Thus we have converted the condition in terms of points drawn uniformly on $S$ to a condition on points that are i.i.d. Gaussian distributed.

Let $X = \sum_{i=1}^{d} (\alpha_i - 1) u_i^2$. In the case that $d$ is sufficiently large we may use the central limit theorem conclude that $X \sim N(\mu, \sigma^2)$. In this regime $X$ will be determined exclusively by its first two moments. We can work out each of these separately,

$$\mu = \mathbb{E}[X] = \sum_{i=1}^{d} (\alpha_i - 1) \tag{8}$$

$$\sigma^2 = \text{Var}[X] = 2 \sum_{i=1}^{d} (\alpha_i - 1)^2. \tag{9}$$

It follows that,

$$P(X > 0) = P(\sigma Z + \mu > 0) = P\left(Z > -\frac{\mu}{\sigma}\right) = 1 - \Phi\left(-\frac{\mu}{\sigma}\right) \tag{10}$$

which proves the result.

