# OpenReview forum: "Adversarial Spheres"
_ICLR.cc/2018/Conference — Invite to Workshop Track_

### Official Review · AnonReviewer2 · 2017-11-27
**Proposes and analyzes one very simple artificial data set, looking for insights about adversarial examples; Despite some good motivations, the significance of the results is not clearly established.**

**Rating:** 4
**Confidence:** 4

**Review:**

The idea of analyzing a simple synthetic data set to get insights into open issues about adversarial examples has merit. However, the results reported here are not sufficiently significant for ICLR.

The authors make a big deal throughout the paper about how close to training data the adversarial examples they can find on the data manifold are. E.g.: “Despite being extremely rare, these misclassifications appear close to randomly sampled points on the sphere.”  They report mean distance to nearest errors on the data manifold is 0.18 whereas mean distance between two random points on inner sphere is 1.41. However, distance between two random points on the sphere is not the right comparison. The mean distance between random nearest neighbors from the training samples would be much more appropriate.

They also stress in the Conclusions their Conjecture 5.1 that under some assumptions “the average distance to nearest error may decrease on the order of O(1 / d) as the input dimension grows large.” However, earlier they admitted that “Whether or not a similar conjecture holds for image manifolds is unclear and should be investigated in future work.” So, the practical significance of this conjecture is unclear.  Furthermore, it is well known that in high dimensions, the distances between pairs of training samples tends towards a large constant (e.g. making nearest neighbor search using triangular inequality pruning infeasible), so extreme care much be taken to not over generalize any results from these sorts of synthetic high dimensional experiments.

Authors note that for higher dimensional spheres, adversarial examples on the manifold (sphere shell) could found, but not smaller d:  “In our experiments the highest dimension we were able to train the ReLU net without adversarial examples seems to be around d = 60.”  Yet,in their later statement in that same paragraph  “We did not investigate if larger networks will work for larger d.”, it is unclear what is meant by “will work”; because, presumably, larger networks (with more weights) would be HARDER to avoid adversarial examples being found on the data manifold, so larger networks should be less likely “to work”, if “work” means avoid adversarial examples.  In any case, their apparent use of only h=1000 unit networks (for both ReLU and quadratic cases) is disappointing, because it is not clear whether the phenomena observed would be qualitatively similar for different fully-separable discriminants (e.g. different h values with different regularization costs even if all such networks had zero classification errors).

The authors repeat the following exact same phrase in both the Introduction and the Conclusion:
“Our results highlight the fact that the epsilon norm ball adversarial examples often studied in defence papers are not the real problem but are rather a tractable research problem. “
But it is not clear exactly what the authors meant by this. Also, the term “epsilon norm ball” is not commonly used in adversarial literature, and the only reference to such papers is Madry et al, (2017), which is only on ArXiv and not widely known — if these types of adversarial examples are “often studied” as claimed, there should be other / more established references to cite here.

In short, this work addresses the important problem of better understanding adversarial examples, but the simple setup has a higher burden to establish significance, which this paper as written has not met.

---

### Official Review · AnonReviewer1 · 2017-11-27
**Exploration of data perturbations in the synthetic problem of classifying 2 concentric spheres**

**Rating:** 5
**Confidence:** 3

**Review:**

The paper considers the synthetic problem setting of classifying two concentric high dimensional spheres and the worst case behavior of neural networks on this task, in the hope to gain insights about the vulnerability of deep networks to adversarial examples. The problem dimension is varied along with the class separation in order to control the difficulty of the problem.

Considering representative synthetic problems is a good idea, but it is not clear to me why this particular choice is useful for the purpose.

2 kind is "attacks are generated" for this purpose, and the ReLU network is simplified to a single layer network with quadratic nonlinearity. This gives an ellipsoid decision boundary around the origin. It is observed that words case and average case empirical error estimates diverge when the input is high dimensional. A Gaussian tail bound is then used to estimate error rates analytically for this special case. It is conjectured that the observed behaviour has to do with high dimensional geometrie.

This is a very interesting conjecture, however unfortunately it is not studied further. Some empirical observations are made, but it is not discussed whether what is observed is surprising in any way, or just as expected? For instance that there is nearly no error when trying to categorise the two concentric spheres without adversarial examples seems to me expected, since there is a considerable margin between the classes. The results are presented in  rather descriptive rather than a quantitative way.

Overall, this works seems somewhat too preliminary at this stage.

---

### Official Review · AnonReviewer3 · 2017-12-01
**List of observations without insights.**

**Rating:** 3
**Confidence:** 3

**Review:**

Adversarial example is studied on one synthetic data.
A neural networks classifier is trained on this synthetic data.
Average distances and norms of errorneous perturbations are computed.
It is observed that small perturbation (chosen in a right direction) is sufficient to cause misclassification.

CONS:
The writing is bad and hard to follow, with typos: for example what is a period just before section 3.1 for? Another example is "Red lines indicate the range of needed for perfect classification", which does not make sense. Yet another example is the period at the end of Proposition 4.1.  Another example is "One counter-intuitive property of adversarial examples is it that nearly ".

It looks as if the paper was written in a hurry, and it shows in the writing.

At the beginning of Section 3, Figure 1 is discussed. It points out that there exists adversarial directions that are very bad. But I don't see how it is relevant to adversarial examples. If one was interested in studying adversarial examples, then one would have done the following. Under the setting of Figure 1, pick a test data randomly from the distribution (and one of the classes), and find an adversarial direction

I do not see how Section 3.1 fits in with other parts of the paper. Is it related to any experiment? Why it defining a manifold attack?

Putting a "conjecture" on a paper has to be accompanied by the depth of the insight that brought the conjecture. Having an unjustified conjecture 5.1 would poison the field of adversarial examples, and it must be removed.

This paper is a list of experiments and observations, that are not coherent and does not give much insight into the topics of "adversarial examples". The only main messages are that on ONE synthetic dataset, random perturbation does not cause misclassification and targeted classification can cause misclassification. And, expected loss is good while worst-case loss is bad. This, in my opinion, is not enough to be published at a conference.

---

### Author Response · Authors · 2018-01-05
**Major changes, and clarifying the significance of the work**

	Thank you to the reviewers for their comments. We have made significant changes to the abstract, and sections 5 and 6 of the paper. We hope the reviewers can reread these sections as it should answer many of their questions as to what insights and takeaways our experiments give. We also clarify in this rebuttal what this paper achieves, what insights can be gained, and why it is significant.
	The goal of this paper was to understand why machine learning models trained on high dimensional spaces are vulnerable to small perturbations (as in the case of image models). Past work has treated the existence of adversarial examples as a problem due to model architecture, loss function, or training data. We consider instead the possibility that even a small amount of classification error may sometimes logically force the existence of many adversarial examples. We use this synthetic dataset to illustrate and explore this phenomenon. Obtaining a complete picture of the geometry of machine learning decision boundaries in high dimensional spaces is extremely difficult and necessitates simplifying the problem where it can be easily understood. The motivation to study concentric spheres is to isolate the effect of high dimensionality of a data manifold on the nature of adversarial examples in a simple setting that can be analyzed theoretically.
	We have updated the paper with a proof of Conjecture 5.1 (now Theorem 5.1). The theorem follows from the special case we proved in the Appendix and an isoperimetric inequality of [1]. Intuitively, what [1] showed is that the subset E of the sphere of a given measure which maximizes the average distance to E is a “pole” (for example in d=3 a local region near the north pole of area 1% is the region which maximizes the average distance to E). What we calculated in the Appendix is that an error set like the north pole of fixed measure will extend to within O(1/sqrt{d}) of the equator, this calculation combined with the result in [1] implies Theorem 5.1. This gives a theoretically optimal tradeoff between the amount of classification error and the average distance to nearest error, and illustrates the counterintuitive nature of high dimensional spaces.
	Amazingly, when we compare several trained neural networks to this optimal bound we see that they naturally are within a small factor of it. This occurs for 3 different architectures, the ReLU and quadratic networks studied in the first version of the paper, and also a large h=2000 depth 8 ReLU network. This behavior can be seen in the updated paper (Figure 5). This answers reviewer 3’s question about larger networks being more vulnerable to small adversarial perturbations. We see that this is not the case at all in Figure 5 - across all 3 architectures, what determines the average distance to nearest error is the accuracy of the network, not the size or complexity of the architecture. Figure 5 demonstrates quantitatively that the model’s decision boundaries are remarkably well behaved, the geometry of the error sets are close to optimal given the amount of test error each network has. Thus the reason random points are “close” to an error is a fundamental consequence of the high dimensional space, and not the result of some other issue of neural networks.
	This paper raises a lot of questions as to the nature of adversarial examples, in particular what is the difference between an adversarial example and a test error? This paper shows that, at least for this dataset, there is no difference between a test error and local adversarial error. Thus there isn’t something “special” to do in order to significantly increase adversarial robustness other than reduce the amount of test error. One type of defense strategy assumes that adversarial examples are off the data manifold, this is what motivated the manifold attack. Several recent submissions to ICLR [2], [3], [4] propose defenses based on this assumption. The manifold attack searches for local errors in the data distribution, and is used to construct all adversarial examples found in this paper. At least in this simple setting we can say confidently that there are local errors both on and off the data manifold, and which ones the attacker finds depends on the attack algorithm.
	Of course this is a toy problem, and we should be cautious before arriving at similar conclusions for image datasets. However, we believe this is an important direction to pursue which completely rethinks the nature of adversarial examples. Perhaps local adversarial errors are a naturally occurring phenomenon on high dimensional datasets? We hope that a complete understanding of this very simple case will point the way forward to explore how the geometry of complex real-world data sets leads to adversarial examples.

[1] Tadeusz Figiel, Joram Lindenstrauss, and Vitali D Milman. The dimension of almost spherical
sections of convex bodies. Acta Mathematica, 139(1):53–94, 1977.

---

> ### Author Response · Authors · 2018-01-09
> **Adding missing references**
>
> [2] Anonymous. The Manifold Assumption and Defenses Against Adversarial Perturbations. https://openreview.net/forum?id=Hk-FlMbAZ
> [3] Anonymous. Defense-gan: Protecting classifiers against adversarial attacks using generative models. https://openreview.net/forum?id=BkJ3ibb0-
> [4] Anonymous. Pixeldefend: Leveraging generative models to understand and defend against adversarial examples. https://openreview.net/forum?id=rJUYGxbCW.

---

### Decision · Program_Chairs · 2018-01-29
**ICLR 2018 Conference Acceptance Decision**

**Decision:**

Invite to Workshop Track

**Comment:**

This paper studies the interplay between adversarial examples and generalization in the uniform setting (not specific assumptions on the architecture) in a toy high-dimensional setting. In particular, the authors show a fundamental tradeoff between generalization error and the average distance of adversarial examples.

Reviewers were skeptical about the possible significance of this work, but the paper underwent a major revision that greatly improved the quality of presentation. That said, the results are still preliminary since they only consider a toy dataset (concentric spheres). The AC recommends re-submitting this work to the workshop track.